# Requirements for Automotive LiDAR Systems

**DOI:** 10.3390/s22197532

**Published:** 2022-10-04

**Authors:** Zhuoqun Dai, Alexander Wolf, Peer-Phillip Ley, Tobias Glück, Max Caspar Sundermeier, Roland Lachmayer

**Affiliations:** 1Institute of Product Development, Leibniz University Hannover, 30823 Garbsen, Germany; 2Cluster of Excellence PhoenixD (Photonics, Optics, and Engineering-Innovation across Disciplines), Gottfried Wilhelm Leibniz Universität Hannover, Welfengarten 1A, 30167 Hannover, Germany

**Keywords:** light detection and ranging (LiDAR), requirements, advanced driver assistance system, autonomous driving, laser scanning, laser safety, object detection, PointPillars

## Abstract

Light detection and ranging (LiDAR) are fundamental sensors that help driving tasks for autonomous driving at various levels. Commercially available systems come in different specialized design schemes and involve plenty of specifications. In the literature, there are insufficient representations of the technical requirements for LiDAR systems in the automotive context, such as range, detection quality, resolving power, field of view, and eye safety. For this reason, the requirements above require to be derived based on ADAS functions. The requirements for various key LiDAR metrics, including detection range, field of view, angular resolution, and laser safety, are analyzed in this paper. LiDAR systems are available with various radiation patterns that significantly impact on detection range. Therefore, the detection range under various radiation patterns is firstly investigated in this paper. Based on ADAS functions, the required detection range and field of view for LiDAR systems are examined, taking into account various travel speeds to avoid collision and the coverage of the entire lane width. Furthermore, the angular resolution limits are obtained utilizing the KITTI dataset and exemplary 3D detection algorithms. Finally, the maximum detection ranges for the different radiation patterns are compared under the consideration of derived requirements and laser safety.

## 1. Introduction

Researchers and manufacturers have rapidly developed light detection and ranging (LiDAR) systems to help perform driving tasks for human drivers and autonomous driving due to their ability to accurately measure distances and detect objects nearly regardless of the lighting conditions [1]. LiDAR systems obtain the distance information by measuring the time difference between emitting and receiving optical signals. The system can be classified as mechanical scanning with a collimated beam or flash irradiation setups with a diverging irradiance [2,3,4,5]. The specifications of some metrics for LiDAR systems, such as detection range, field of view (FOV), and angular resolution, are suggested in many studies [6,7,8,9,10,11,12,13,14]. However, a consistent result cannot be summarized from them. Since the respective solution depends on the boundary conditions and requirements, a detailed consideration of these influencing factors is necessary. For example, LiDAR systems that use collimated beams can acquire a long detection range due to the small energy drop off from the emitter to the target. However, the collimated beam must be steered with a significant pulse repetition rate, limiting the emitted energy concerning laser safety regulations [15]. This paper evaluates the detection range for various beam radiation patterns, including flash irradiation and various scanning approaches. The criteria for the detection range, FOV, and the angular resolution are determined under the consideration of ADAS functions, serving as a benchmark for the specification and design of LiDAR systems.

## 2. State of the Art

The detection range, FOV, and angular resolution are essential metrics that affect the performance of LiDAR systems. This chapter introduces the current state of these metrics and the functional principle of LiDAR systems.

### 2.1. Current Requirements for Automotive LiDAR Systems

So far, the performance requirements of automotive LiDAR are not unified. The study by Warren [6] demonstrates the requirements for automotive LiDAR systems regarding discussions with manufacturers, wherein LiDAR systems are defined as short-range from 20 m to 30 m and long-range up to 300 m, individually. The angular resolution for short-range LiDAR systems is proposed at 1°, while the long-range LiDAR systems require an angular resolution between 0.1° and 0.15° to detail remote objects. Zhao et al. [9] and Yeong et al. [10] list the current best products of various manufacturers in terms of performance metrics in their study, including detection range, FOV, and angular resolution. The statistics indicate that commercial LiDAR systems have a maximum detection range of up to 300 m concerning a reflectivity of 10% of the detection object. A horizontal FOV of 360° appears only in mechanical LiDAR systems, while other systems cover a horizontal view angle between 60° and 270°. The vertical FOV varies between 4° and 60°, and a finest angular resolution of 0.1° in both directions can be summarized from commercial products. In the study by Takashima et al. [13], the specifications for LiDAR systems are listed for an advanced driver assistance system (ADAS) and robotic car applications based on the sampling rate of current mechanical scanning LiDAR systems. In their study, forward-looking LiDAR systems have to detect objects up to 200 m with a 14° × 3.5° FOV and 0.3° × 0.3° angular resolution. In contrast, side looking LiDAR systems are required to have a 30 m range with a 60° × 5.7° FOV and 0.2° × 0.2° angular resolution. Raj et al. state that LiDAR systems for ADAS applications require a minimum detection range between 100 m and 200 m and a horizontal FOV of more than 90° [14]. The studies introduced above indicate that it is difficult to summarize a particular standard of LiDAR metrics. Additionally, the particular application scenarios must be taken into account while determining the requirements. For instance, the angular resolution affects the density of the point cloud, which also affects the accuracy of the object detection via LiDAR systems.

### 2.2. Functional Principle and Principal Components of LiDAR Systems

LiDAR systems obtain the range information based on the time of flight (TOF) of an emitted light signal approach. The distance between a LiDAR system and a target is computed by half the multiplying of the light velocity and the signal’s flight time when neglecting the distance between the emitter and detector. The TOF refers to the time difference or phase shift between the emitted and received signals, by which emitted beams are modulated as pulsed or as continuous waves (CW).

In pulsed LiDAR systems, the emitter radiates light pulses with a nanoseconds duration [14]. The TOF is directly estimated using the time difference between pulse transmission and reception. This approach allows LiDAR systems to emit a high instantaneous power to obtain a long detection range concerning the limit of eye safety.

The continuous wave amplitude modulated (AMCW) approach regulates the intensity of a continuous wave with a constant frequency $f_{A}$. The object distance R is determined using the phase shift between emitted and received waves ∆φ and the light speed c with:(1)$$R=\frac{c\cdot ∆\varphi }{4\cdot \pi \cdot f_{A}}$$

The AMCW method is often applied in TOF cameras for indoor applications, involving cost-effective LEDs as emitters rather than laser diodes [16]. The available power restriction prompts a low signal-to-noise ratio (SNR) and, consequently, a short detection range [16]. In addition, as the object distance increases, AMCW LiDAR systems cannot receive enough photons to determine the phase difference, which also restricts their applications for long-range detections.

Besides the AMCM method, the emitter signal can also be modulated with a periodically shifted frequency in a continuous wave frequency modulated (FMCW) approach. The received signal wave is blended with the emitted wave as references, leading to a frequency difference $∆f_{F}$. The object distance is obtained under the consideration of the bandwidth B and the period of the frequency modulation T with:(2)$$R=\frac{∆f_{F}\cdot c\cdot T}{2\cdot B}$$

Waves must interfere with themselves at different times to distinguish the frequency differences. Hence, the laser coherence must be maintained in the round trip to the target to determine the frequency difference [17]. Commercially available diode lasers regularly have a coherence length of <1 mm [18]. In contrast, external cavity diode lasers (ECDL) and fiber lasers can be utilized as emitters to achieve an over 100 m detection range [18]. However, external resonators used to stabilize the coherence length improve the size and cost of such LiDAR systems.

As emitters, diode lasers are preferred in LiDAR systems for automotive applications due to their compact size and minimal expense. The two most commonly used diode lasers are edge-emitting lasers (EEL) and vertical-cavity surface-emitting lasers (VCSEL). Compared with the EELs, VCSELs have a minor spectral shift in terms of temperature (approximately $0.06\;\mathrm{nm}\cdot K^{-1}$ versus $0.25\;\mathrm{nm}\cdot K^{-1}$) [17]. Thus, using a narrow bandpass filter can achieve an improved SNR. Besides this, VCSELs can be fabricated in an array so that multiple laser sources can be used in one compact system. In this way, the pulse repetition rate of each laser can be reduced, increasing the detection range, which is limited by laser safety aspects which are discussed in Section 4.4.

In order to detect reflected photons regularly, either linear mode avalanche photodiodes (APDs) or single-photon avalanche diodes (SPADs) are utilized in LiDAR systems. Linear-mode APDs produce a signal amplification with a typical gain of ca. 200, proportional to the incoming light [16]. However, they are not able to detect a single photon. In contrast, SPADs are APDs operating in Geiger mode with a gain greater than 10^4^ once it is stimulated with even a single photon [16]. Hence, it can be used when an extremely high sensitivity for single-photon detection is required. However, one disadvantage of SPAD is the dead time of 400 ns to 1 µs between two adjacent measurements [19]. During this time, the detector is saturated and other photons received by the detector no longer generate an electrical signal. The pulse repetition rate is thus limited. Due to the high sensitivity, the sensor is easily affected by noise. Therefore, for LiDAR systems using SPAD sensors, one frame image usually requires 50 to 100 measurements per pixel to reduce the SNR and increase the detection accuracy [20].

As stated above, the detection range is one of the essential metrics for LiDAR systems. AMCW LiDAR systems require a high-power emitter to capture sufficient reflected photons to determine the phase difference. However, the high emitted power can lead to eye hazards concerning laser safety. For FMCW LiDAR systems, the coherence length of a cost-effective laser source limits their maximum detection range. In pulsed LiDAR systems, short pulse widths and low-duty cycles can help achieve a high peak emitted power to increase the detection range, limiting the exposure to ensure the eye-safe. Therefore, they are appropriate for automotive applications and are focused on in this paper.

## 3. Range Equation for Different Radiation Patterns

Different beam shaping methods are available in LiDAR systems, producing various radiation patterns. This chapter proposes a range equation for LiDAR systems and evaluates the detection range of LiDAR systems with various beam patterns.

According to the usual definition, the detection range of LiDAR systems is the distance at which the reflection of an object with 10% reflectivity and a Lambertian backscatter characteristic can still be detected by a detector pixel [21]. Equation (3) describes the relationship between the emitted power $P_{s}$ and received power $P_{r}$ of LiDAR systems. The main variables in Equation (3) are listed in Table 1.
(3)$$P_{r}=\frac{P_{s}T_{A}\eta_{t}}{A_{s}}\cdot \Gamma \rho \cdot \frac{T_{A}}{{\pi r}^{2}}\cdot A_{R}\eta_{r}$$

Among the Equation (3), $A_{s}$ is the beam spread area of the emitter at the target. $T_{A}$ and $\eta_{t}$ are the atmospheric transmission and the optical efficiency of the emitter. The first term of Equation (3) indicates the optical irraidance (W∙m^−2^) incident on the target. In the second term, the target cross-section Γ is the irradiated area of the target. ρ is the reflectance of the target. The product of the first and the second term is equivalent to the total optical power scattered by the target in the detector’s direction. In the third term, r is the distance between the LiDAR system and the object. An omnidirectional (4π) scattering is assumed for microwave radar systems that operate a millimeter-wavelength for small objects. However, for LiDAR systems that operate with shorter wavelengths (near-infrared), the scattering only occurs in the backward direction. The maximal radiant intensity is normal along the surface and 1/π of the total reflection radiant intensity, assuming Lambertian scattering (see Appendix A). The product of the first three terms gives the maximal optical irradiance of the backscatter at the receiver. After multiplying the optical aperture $A_{R}$ and the optical efficiency $\eta_{r}$ of the detector, the optical power received by the detector can be obtained.

In Equation (3), the beam spread area $A_{s}$ is a function of the distance r and the radiation angle in horizontal ($\varphi_{h}$) and vertical ($\varphi_{v}$) directions. The cross-section Γ is related to the irradiated area viewed by a single detector pixel, which is a function of the distance r and the opening angle of a single pixel α. Therefore, the ratio of the cross-section Γ and the beam spread area $A_{s}$ must be further investigated for different radiation patterns with different radiation angle. Accordingly, the Equation (3) can be simplified with:(4)$$P_{r}{=P}_{s}\cdot K\cdot \frac{\Gamma }{A_{s}}$$
where the factor K is the set of the parameter irrelevant to the beam patterns and given with:(5)$$K=\frac{T_{A}{}^{2}\cdot \eta_{t}\cdot \eta_{r}\cdot A_{R}\cdot \rho }{{\pi r}^{2}}$$

Figure 1 shows three radiation patterns depending on the beam shaping relative to the size of a quadratic-shaped single-pixel image.

In the first case, the laser beam is assumed to be collimated with a circular or elliptical cross-section. Either a single-pixel detector or a 2D array detector can be used for this case. The beam spread area $A_{s}$ is smaller than the area viewed by a detector pixel (black rectangle in Figure 1a) for all relevant distances. Therefore, the cross-section Γ equals to the beam spread area $A_{s}$.

In the case of LiDAR systems using blade irradiation, as shown in Figure 1b, the total irradiated area is larger than the a pixel image only in one direction. The scanning direction can be horizontal or vertical to cover the entire FOV. In this case, either a 1D linear detector array or a 2D detector array can be used. According to the schematic, the cross-section Γ of a single-pixel occupies only a small part of the beam spread $A_{s}$, determined by the width ratio between the image size of a single pixel and the total number of pixels. Due to this ($\alpha \;\ll {\;\varphi }_{h}$), the received energy $P_{r}$ decreases significantly compared to the spot irradiation concerning a constant emitted optical power $P_{s}$ and factor K. For vertical blade irradiations, the indices h and v have to be swapped in the corresponding equations of Table 2.

Since the entire FOV is irradiated at once in flash LiDAR systems, only a 2D array detector can be used to generate the required resolution. The cross-section Γ is determined by the size of a pixel image on the irradiated surface (a black unit in Figure 1c), which is much smaller than the beam spread area $A_{s}$, leading to a further decrement of received energy $P_{r}$ compared to the other two cases. The range equation for all cases is summarized in Table 2. The derivation of the beam spread area $A_{s}$ and the cross-section Γ for all cases is given in Appendix A.

For the discussion above, we focus only the center pixel and the object that is directly in front of the LiDAR system. In this case, the maximum reflected radiant flux can be received on the pixel when Lambert backscatter is considered. In addition, it is assumed that a target shows at least one point in LiDAR frames, leading to the target size being larger than the size of a single pixel image. Some special cases are neglected such as a small target appearing in a long distance. LiDAR systems detect only one point, which makes object detection difficult. Among the three equations, the irradiance angle of the emitter ($\varphi_{h}$ and $\varphi_{v}$), the angle viewed by a single-pixel (α), and the detection range (r) play essential roles to describe the energy flow of LiDAR systems. Moreover, these parameters have significant impacts on the design metrics of the FOV, angular resolution, and the detection range. Therefore, their requirements have to be investigated in the following sections.

## 4. Investigation of the Requirements for LiDAR Systems

Commercially available LiDAR products show a variety of concepts and capabilities. So far, most studies on LiDAR sensors barely focus on discussing the requirements. In contrast, automotive lighting is regimented in detail by standards and regulations [22,23,24,25,26,27,28]. This chapter investigates the necessary detection range, field of view, and angular resolution of LiDAR systems, considering advanced driver assistance system (ADAS) functions and object detection via LiDAR point clouds.

### 4.1. Detection Range of the ADAS Applications

ADAS support the driver by assisting in longitudinal and lateral driving tasks, e.g., braking and steering, to increase safety and comfort while driving. Adaptive cruise control (ACC) is a typical ADAS function that adjusts the subject vehicle’s speed to ensure a safe distance from a preceding vehicle. Another important ADAS function that focuses on longitudinal driving tasks is the forward vehicle collision warning system (FVCWS), which monitors the speed and distance of the vehicle driving ahead to prevent a collision, or at least reduce its severity. While detecting a non-moving vehicle is optional according to ISO 15623 [29], static obstacles discussion is included in this paper.

Since LiDAR systems provide accurate distance information, they are mainly used for ACC and FVCWS applications [30,31]. Because forward detection often requires a more extended range than monitoring the surrounding environment, the following discussions of the requirement are based on the assumption that an individual sensor located in the middle of the vehicle front is used.

The function of ACC is to control the driving speed to adapt to a preceding vehicle [32]. The maximum required detection range $d_{\max}$ of the ACC function is calculated with the maximum driver-selectable set speed $v_{\max}$ and the maximum selectable time gap $\tau_{\max}$ by [32]:(6)$$d_{\max}{=\tau }_{\max}\cdot v_{\max}$$

The standard ISO 15622 [32] specifies that the minimum selectable time gap for the subsequent control shall be ≥0.8 s, and the system shall provide at least one time gap between 1.5 s and 2.2 s for speeds higher than 8 m·s^−1^ (30 km·h^−1^). For example, a detection distance of 85 m is required to have a time gap of 2.2 s at a speed of 38.8 m·s^−1^ (140 km·h^−1^).

The function of the FVCWS is to warn the driver when a preceding vehicle appears in the trajectory of the subject vehicle and becomes a potential hazard [29]. The standard ISO 15623 [29] specifies the range requirements regarding scenarios where a warning rises. Among them, three typical scenarios for the FVCWS according to different states of the obstacle vehicle are indicated in Table 3.

In the first scenario, the preceding vehicle is traveling at the same speed as the subject vehicle, where the warning distance is only related to the subject vehicle’s speed $v_{1}$ and the driver’s brake reaction time T.

In the second scenario, the preceding vehicle is regarded as a stationary obstacle. The subject vehicle’s deceleration $a_{1}$ must be taken into account by determining the warning distance.

In the last scenario, the preceding vehicle is decelerating with a relative speed $v_{\mathrm{rel}}$ relative to the subject vehicle, while the subject vehicle is traveling with $v_{1}$. Scenario one and scenario 2 are the two boundary conditions of scenario 3 when $v_{\mathrm{rel}}=0$ and $v_{\mathrm{rel}}=v_{1}$, respectively.

Increasing the brake reaction time T and the subject’s vehicle deceleration $a_{1}$ increase the required warning distance. ISO 15623 utilizes the result in the study of Johansson et al. [33] to determine the brake reaction time T. The study indicates that the reaction time of 98% of tested people is under 1.5 s. Moreover, the minimum subject vehicle deceleration $a_{1}$ is given with 5.3 m·s^−2^ by evaluating the emergency brake performance of cars on a dry, flat road surface in ISO 15623 [29]. Different braking capabilities have to be additionally considered for other vehicles like trucks. The maximum warning distances for the three scenarios are presented in Figure 2.

Although the driving speed on several motorways is not limited in Germany, most countries regulate the maximum driving speed. Typical speed limits are between 90 km·h^−^^1^ and 130 km·h^−^^1^. According to [34], the maximum limit of the listed countries is 140 km·h^−^^1^, chosen as the subject vehicle speed for scenario 3 in Figure 2. Scenario 2 is a boundary condition of scenario 3 when the preceding vehicle decelerates until it stops. Therefore, concerning all scenarios and a subject vehicle speed of 140 km·h^−^^1^, a minimum detection range of 200 m is required to avoid a forward collision.

Noticeably, overtaking maneuvers on rural roads require an extensive detection range to avoid a forward collision when the oncoming traffic is considered. For example, the German rural road is classified into four types [35]. Two of them are designed with overtaking lanes, which means the overtaking maneuver can be achieved without the influence of the oncoming traffic. The other two rural roads have a maximum design speed of 90 km·h^−1^ without overtaking lanes. Assuming that both the subject and oncoming vehicles are traveling at 100 km·h^−1^ (which is the legal maximum and more applicable than the reduced design speed), a warning distance of 180 m is required.

### 4.2. Field of View Requirements for ADAS

Compared with straight roads, detection in curves requires an increased horizontal FOV. The ACC system shall enable the subject vehicle to follow a preceding vehicle on curves with a minimum radius of 500 m [32]. The geometric relations to cover the complete lane width in a curve are shown in Figure 3a. Equation (7) can be used to determine the half horizontal viewing angle θ to completely cover the width of the own lane up to the apex of the inner curve, assuming that the subject vehicle’s position is in the middle of the lane. A larger horizontal angle is required to capture more distance in a curve.
(7)$$\theta =\arccos(\frac{{R-W}_{L}/2}{R})$$

In Equation (7), R is the curve radius, and $W_{L}$ is the lane width. The lane width typically varies between 2.75 m and 3.75 m depending on the design speed and the number of parallel lanes [35,36]. Based on the maximum value of 3.75 m, Figure 3b shows the required whole horizontal angle to cover the complete lane width as a function of the curve radius. Accordingly, the minimum horizontal FOV required for the ACC function is approx. ±5° concerning a 500 m curve radius.

The forward vehicle collision warning system (FVCWS) function is classified by curve radius capability that an obstacle can be detected in a curve with a certain radius in the subject vehicle’s trajectory. System classes Ⅰ, Ⅱ, and Ⅲ refer to the curve radii of >500 m, >250 m, and >125 m, respectively. Detecting a preceding vehicle with a lateral offset of up to 20% of its width must be possible from the minimum distance, which varies between 5 m and 10 m depending on the system class [29]. Figure 4 illustrates these geometric requirements.

The minimum detection height $h_{1}$ and the maximum height h from the ground are given in [29] with 0.2 m and 1.1 m, respectively, to determine the vertical FOV for the FVCWS function. Concerning a vehicle width $W_{v}$ of 2 m [37], the resulting FOVs for the FVCWS function are listed in Table 4.

The results in Table 4 indicate that a smaller curve radius requires a larger FOV in horizontal and vertical views to detect potentially dangerous objects in the current lane. Concerning a minimum curve radius of 125 m, the horizontal angle to cover the full lane width must be more than 20° (see Figure 3b), which is below the Class Ⅲ FVCWS. In summary, ADAS functions that focus on principal longitudinal driving tasks require a FOV of 32.6° × 10.2° concerning curve radii ≥ 125 m.

### 4.3. Angular Resolution Requirements

The angular resolution is defined by the minimum angular distance between two objects which can be resolved by LiDAR systems [31]. According to the description in chapter 3, the angular resolution refers to the angle of two adjacent scanning points for scanning LiDAR systems and two adjacent detector pixels for non-scanning LiDAR systems.

Angular resolutions of 0.1°~0.2° are stated for LiDAR systems [9,10,11,12,13]. Since it directly affects the detection performance, object detection algorithms based on LiDAR point clouds must be considered when determining the required angular resolution.

Table 5 compares the average precision and processing time for commonly used open-source LiDAR-based object detection algorithms. The algorithms are trained and tested on the KITTI validation set [38] via OpenPCDet toolbox, using RTX 3080 GPU and i7-10700K CPU.

The KITTI dataset contains 7481 LiDAR frames (images) for training and 7518 LiDAR frames for testing. Among them, 80,256 objects are labeled, including eight categories, e.g., ‘Car’, ‘Truck’, ‘Pedestrian’, and ’Bicyclist’. The statistic of the dataset shows that cars, pedestrians, and bicyclists are the most predominant categories [38]. Hence, they are focused on in this paper.

The average precision shown in Table 5 is evaluated using the PASCAL metrics [43], resulting from all the 7518 testing LiDAR frames in the KITTI dataset. In object detection tasks, recall is defined as the proportion of detected relevant elements over the total number of the relevant elements. Precision is the proportion of detected relevant elements among all detected elements. The average precision is computed from the precision/recall curve and defined as the mean precision at 40 equally spaced recall levels.

According to the results listed in Table 5, each algorithm has minor differences in the detection performance of traffic objects. The average precision of all algorithms for identifying cars, pedestrians, and bicyclists has a standard deviation of 0.59%, 3.95%, and 1.77%, respectively. The PointPillars algorithm requires the shortest processing time for each image. Under the hardware conditions above, the processing of each image requires 0.027 s with GPU and 0.0286 s with CPU. This leads to a ~35 Hz data processing rate. An automotive sensor has to deal with real-time dynamic information to capture moving objects or obstacles. A minimum frame rate of 25 Hz has to be considered while selecting the frame rate for an automotive LiDAR [44]. Concerning this requirement, the data processing has to be greater than 25 Hz to achieve real-time detection. PointPillars is the only algorithm that achieves a frame rate of over 25 Hz real-time detection on the tested system. Despite its average precision by pedestrians and bicyclists being slightly lower than other algorithms, considering the importance of real-time detection for automotive applications, it is selected as an example to obtain the angular resolution requirement in this paper.

Classification and position tasks are usually used to evaluate a detection result, e.g., “is there a car in the scenario?” and “where is it?” [45]. The result is often visualized as a bounding box for each detected object with a confidence score.

The confidence score is the product of the probability that an object contained in a bounding box (P_object_) and the intersection over union (IoU). If no object is contained in a box, the confidence score equals zero. Otherwise, it equals the IoU. The IoU indicates the overlap ratio between the predicted box $B_{p}$ and the ground truth $B_{g}$, given with [46]:(8)$$\mathrm{IoU}=\frac{{\mathrm{area}(B}_{p}\cap B_{g})}{{\mathrm{area}(B}_{p}\cup B_{g})}$$

Lang et al. [42] introduce the PointPillars algorithm in their study at first, and the threshold of the IoU is set to 0.5. According to Equation (8), an IoU of 0.5 refers to a maximum 33.3% offset between the predicted box and the ground truth, e.g., in the horizontal direction, leading to a maximum 0.67 m offset of car detections concerning a width of 2 m. The threshold of an IoU can be set to more than 0.5 to obtain a more accurate location prediction. Since many LiDAR-based object detection algorithms and evaluation metrics prefer to apply an IoU threshold of 0.5 to consider a correct detection [39,40,41,45], the angular resolution limits are acquired with 0.5 confidence as a reference in this paper.

Three of the most common road users are chosen for the detection. For cars and bicyclists, the detection is evaluated for two perspectives due to these participants’ different width and depth dimensions. Under an identical distance, the object at a larger cross-section perspective may be easily detected, leading to a minimum angular resolution requirement. Since pedestrians show a similar width and depth, only one perspective is taken into account.

The raw point clouds are downsampled to different angular resolutions to find a limit for the 0.5 confidence score. The Velodyne HDL-64E LiDAR used in the KIITI dataset has an angular resolution of 0.08° in the horizontal and 0.4° in the vertical direction. These values decide the finest possible angular resolution for the image and the minimum interval for downsampling. The upper limits for horizontal and vertical angular resolution are selected at 1.04° and 1.6°, respectively, providing enough tolerance for determining the angular resolution limit. For every object and perspective, 52 downsamplings are executed. The detection results for the three object types are shown in Figure 5.

The right column in Figure 5 shows the confidence score for various angular resolutions for the scenario shown in the corresponding scene in the left column. The threshold of the confidence score is set to 0.4. Since detections with a lower confidence score cannot be treated as positive classifications, they are shown as blank fields in the heatmap. The Velodyne LiDAR has a different angular resolution in the horizontal and vertical directions. Since using quadratic pixels is a more general approach to realize than using rectangular ones, especially with flash LiDAR systems, the required angular resolution for the confidence score of 0.5 is determined using this pixel shape.

The required number of LiDAR points is determined from the calculated results shown in the right column of Table 6. In combination with the aspect ratio of the detection objects, this table gives the required number of quadratic shaped pixels (last column in Table 6). The point numbers leading to a score between 0.45 and 0.55 are averaged. With a maximum variance of 5%, the estimated findings are able to offer a detection confidence of 0.5. The object aspect ratio is determined using the dimension of the raw point cloud shown in the middle column in Figure 5. Since car windows usually do not appear in LiDAR point clouds (they do not backscatter the LiDAR irradiation), only the lower half of the vehicle body is considered.

Due to the different sizes of the objects, bicyclists and pedestrians require a finer angular resolution than cars for a successful detection at the same distance. In addition, objects are more difficult to detect in the rear view than in the side view. The resulting dependency between object distance and required angular resolution is shown in Figure 6.

As shown in Figure 6, the standard deviation of the required angular resolution for all scenarios reduces with an increasing distance. According to the discussion in Section 4.1, a minimum range of 200 m is able to prevent a collision for traveling speeds of up to 140 km∙h^−1^. This distance requires an average angular resolution of 0.07° with a standard deviation of 1.8% for all scenarios. The highest criterion among them is bicyclist detection in the rear perspective view. An angular resolution of less than 0.04° is required. This value will be considered for evaluating laser safety in the following section.

### 4.4. Laser Safety and Comparison of the Detection Range

Besides the influence of the radiation pattern, the output power, which is limited by eye safety, strongly affects the detection range. According to the IEC 60825-1 standard, automotive LiDAR systems are certified with laser class 1 as safe for the human eye [29]. The following parameters are criteria that influence the determination of the accessible emission limit (AEL) under the consideration of eye safety [15]:Wavelength;Pulse duration;Frame rate;Number of pulses per frame;Laser beam divergence.

The most significant hazard for wavelengths between 400 nm and 1400 nm is thermal damage to the eye’s retina. For pulsed LiDAR systems that operate with a wavelength <1400 nm, the output power limit is determined by applying the most restrictive of these three conditions [15], namely:The maximum AEL for a single pulse (AEL._single_);The average power for a pulse train (AEL._s.p.T_) of an emission duration T;The AEL for a single pulse multiplied by a correction factor C5 (AEL._s.p.train_).

Among them, the emission duration T varies from 10 s to 100 s regarding the divergence angle of the laser beam [15]. The correction factor C5 is determined by the effective number of pulses for a given exposure duration [15]. Only AEL._single_ and AEL._s.p.T_ have to be compared for LiDAR systems that operate with a wavelength > 1400 nm. For the wavelengths > 1400 nm, radiation penetrates into the aqueous humour, where the heating effect is dissipated due to the water absorption [15]. The following parameters are assumed to compare the limitation of the output power for LiDAR systems with different radiation patterns indicated in Section 3:Pixel number: 815 × 255;Frame rate: 30 Hz;Beam divergence in spot scanning LiDAR systems: <1.5 mrad;Single pulse duration: 5 ns;Number of light sources for each pattern: 1.

As discussed in Section 4.2 and Section 4.3, LiDAR systems require a FOV of 32.6° × 10.2° for ADAS functions. It is known that detecting a small object at a long distance requires a finer angular resolution than other situations. To detect a bicyclist with a rear perspective at 200 m, an angular resolution of 0.04° × 0.04° is necessary (see Section 4.3). Accordingly, the pixel number of the considered LiDAR systems is 815 × 255. The pulse repetition rate of flash LiDAR systems is equivalent to the systems frame rate, defined here at 30 Hz.

Meanwhile, blade irradiation scanning LiDAR systems need only to scan in one direction. Hence, the pulse repetition rate of blade irradiation scanning LiDAR systems is 815 × 30 Hz for horizontal and 255 × 30 Hz for vertical scanning. For spot scanning LiDAR systems, the angle between two adjacent scanning points is the angular resolution. Thus, the laser’s beam divergence must be less than the angular resolution. Concerning a 0.04° angular resolution (see Section 5), the beam divergence must be less than 0.70 mrad. Hence, the AEL of the spot scanning LiDAR systems is evaluated according to [15] for beam divergences < 1.5 mrad.

On the contrary, the AEL of blade irradiation scanning and flash LiDAR systems is obtained according to [15] for extended source radiations. Moreover, a typical single pulse duration of 5 ns is assumed [16] to determine AEL._single_. Figure 7 shows the maximum accessible exposure limits as a function of the wavelength for the three radiation patterns concerning the retinal hazard region (wavelength up to 1400 nm).

The AEL values in Figure 7 indicate the energy limitation falling into the eye, assuming a 7 mm pupil aperture and 100 mm measuring distance. The most critical value of the three curves (two curves for wavelength > 1400 nm) has to be considered for a specific wavelength. The pupil aperture only occupies part of the irradiated area, depending on the distance and the beam divergence angle. Hence, the most restricted AEL value is not identical to the emitted energy of LiDAR systems which increases the possible emission of a LiDAR system.

Equation (9) indicates the calculation of the total accessible emitted energy at the pupil, which is donated by $E_{p}$.
(9)$$E_{p}=\frac{\mathrm{AEL}}{A_{r}}$$
where AEL is the energy limitation on the retina from Figure 7. $A_{r}$ is the ratio of the pupil area Ap and the size of the area irradiated by the laser AL in a certain distance. For Ap > AL, all emitted energy may fall into the observer’s eye, leading to $E_{p}=\mathrm{AEL}$.

To further compare the detection range of the three radiation patterns mentioned in Section 3, the following assumptions are made:

Laser wavelength: 905 nm.Laser divergence angle:○spot scanning: 0.04° × 0.04° (equals the resolution requirement);○blade irradiation, horizontal scanning: 0.04° × 10.2°;○blade irradiation, vertical scanning: 32.6° × 0.04°;○flash irradiation: 32.6° × 10.2° (equals the FOV).

Optical efficiency of the emitter: 90%.Optical efficiency of the detector: 90%.Reflectivity of the object: 10% with Lambertian scattering characteristic (Section 3).Aperture of detector optical system: Ø 25.4 mm (1′′).Pupil diameter: Ø 7 mm.Atmospheric attenuation and scattering: neglected.Pixel gap: neglected.Intensity distribution of the emitter: homogeneous, K = 1.

A typical wavelength of 905 nm can be stated for automotive LiDAR systems [1,9,47,48]. Therefore, a wavelength of 905 nm is considered for exemplary systems. According to Figure 7, the emission energy for a spot scanning LiDAR system is limited to 0.1608 nJ at this wavelength. The limit is 77.49 nJ for vertical and 247.7 nJ for horizontal blade irradiations. In contrast, the emission energy of flash LiDAR systems can be up to 445.3 nJ.

The type of detector determines the required receiving energy of LiDAR systems. The comparison of the maximum detection range for different radiation patterns has to be assumed to use identical detectors. For example, SPADs are appropriate for LiDAR systems to achieve a long detection range due to their single-photon sensitivity. The energy of a single-photon saturates the detector, causing it to enter the death time (see Section 2.2). During this death time, other photons received by the detector no longer generate electrical signals. In this case, the required energy for one detection equals the energy of a single photon. Multiple measurements can compensate for the uncertainty caused by single-photon measurements (see Section 2). The energy of a single photon $E_{\mathrm{photon}}$ at 905 nm is 2.2 × 10^−19^ J, according to Equation (10), with h as Plank’s constant and the light speed c.
(10)$$E_{\mathrm{photon}}=\frac{h\cdot c}{\lambda }$$

The nominal ocular hazard distance (NOHD) is the distance from the output aperture, beyond which the emitted energy remains below the AEL [15]. For each assumed NOHD, the corresponding maximum energy output can be calculated according to Equation (9). This leads to a detection range (calculated with the equations in Table 2) for each scanning type depending on the chosen NOHD (Figure 8).

For spot irradiation, a laser beam with a divergence angle of 0.04° extends to the pupil diameter of 7 mm in about 10 m. All the emitted energy is able to fall on the retina up to this range. Hence, the total emitted energy equals the AEL value and leads to a constant detection range of ~103 m. As the NOHD increases further, the spot irradiation covers a larger area than the pupil aperture, allowing an increased energy output and an extensive detection range. Concerning a detection range of 200 m, an NOHD of ~20 m is required.

For blade and flash irradiation, the detection range increases significantly with the NOHD. With the given assumptions, horizontal and vertical blade irradiation enables a detection range of the required 200 m with an NOHD of 0.03 m. These distances can easily be maintained with a suitable housing or mounting position of the system. For flash irradiations, the eye-safety distance (NOHD) is 0.17 m.

## 5. Discussion

In the Section 6, we examine the required detection range and FOV for LiDAR systems based on ADAS functions. Using an exemplary algorithm and different object categories, the requirement of angular resolution is determined by object detection. Furthermore, different beam irradiation patterns are compared for the eye-safety distance (NOHD).

Section 4.1 deduces the required detection range of LiDAR systems based on collision avoidance. Concerning a maximum speed of 140 km∙h^−1^, LiDAR systems must be able to detect objects in a distance of 200 m. In Germany, several motorways have no speed limit. LiDAR systems require a more extensive detection range for this case, and no current system can guarantee driving safety for an unlimited speed scenario.

The discussion of the FOV requirement in Section 4.2 focuses on the coverage of the subject lane, which results in a limited horizontal FOV. Therefore, detecting a cut-in vehicle can be very late, as illustrated in Figure 9a. A substantial offset (sensors capture only a part of the object width) of the traffic can also lead to failures in detection, especially in detecting motorbikes due to their narrow silhouette [49]. A similar situation happens on bicycles riding on the street (Figure 9b). These issues can be compensated by extending the horizontal FOV of the sensors or by using additional sensors with a large horizontal FOV and possibly a short detection range (Figure 9c). With a lane width of 3.75 m and a minimum distance to a cut-in vehicle of 2 m at a low speed [49], an entire horizontal FOV of 140° is required to cover the width of three lanes.

Section 4.3 investigates the angular resolution limits of LiDAR systems based on the PointPillars object detection algorithm. Concerning an angular resolution of 0.1°, as stated in the mentioned literature (see Section 2), a vehicle in the rear view can be detected at a maximum distance of 150 m. The speed of a subject vehicle must be less than 100 km∙h^−^^1^ to avoid forward collisions at this distance. This warning distance allows a traveling speed of 100 km∙h^−1^ to prevent forward collisions. Considering oncoming traffic, the subject and oncoming vehicles must drive at less than 65 km∙h^−1^ (assuming the same speed for both vehicles).

As the driving speed increases or smaller detection objects have to be considered (such as bicycles or motorbikes), this detection distance and the angular resolution are insufficient. For a speed of 100 km∙h^−^^1^ of both approaching vehicles, a detection range of 180 m is required.

Considering a motorway (the detection of oncoming traffic is not relevant) and a driving speed of 140 km∙h^−^^1^, a detection range of 200 m is required. In this case, the minimum angular resolution in the horizontal and vertical directions to identify a stationary passenger car is 0.075°. For smaller objects, like (motor)cycles in the rear view, a minimum angular resolution of 0.04° is required for a confidence score of 0.5. LiDAR systems require sharper lenses and detectors with more pixels to obtain a superior angular resolution, which can raise the manufacturing cost. Instead, improving the accuracy of neural networks by using such type-3 fuzzy logic systems [50] can increase the detection precision in low-density point clouds, while decreasing the requirements of the angular resolution for LiDAR systems.

Section 4.4 compares the maximum detection range for three radiation patterns concerning laser safety limitations. With blade and flash irradiation, the required detection range of 200 m is realized with a safety distance (NOHD) of less than 0.2 m. Furthermore, for the very low NOHD of 0.03 m of the blade irradiation in the example presented in this paper, a mechanical solution like an appropriate housing or mounting position of the LiDAR system is sufficient to ensure eye safety.

LiDAR systems that scan with a collimated laser spot have to meet the demand of a significant pulse repetition rate, significantly limiting the output power to maintain eye safety. To increase the detection range, the emitter can be designed as an array of lasers to decrease the pulse repetition rate when using a single laser diode.

Another possibility for increasing eye safety or the range of LiDAR systems is using lasers with longer wavelengths, for example, 1550 nm. In this way, a significant detection range can be achieved even with the technologically simple approach of flash LiDAR while maintaining eye safety. The tradeoff, however, is the required use of an InGaAs-based detector, which is significantly more expensive than a conventional one for lower wavelengths.

## 6. Conclusions

In this paper, the requirements of essential LiDAR metrics are investigated. Concerning a motorway speed limit of 140 km∙h^−^^1^, LiDAR systems require a minimum detection range of 200 m to avoid the worst case of forwarding collisions. Regarding the ACC and FVCWS function, LiDAR systems must have a FOV of 32.6° × 10.2° to detect preceding vehicles in curves with a minimum radius of 125 m. The angular resolution requirements are obtained based on the PointPillars algorithm with a confidence score of 0.5. A LiDAR system must have an angular resolution of 0.04° × 0.04° to classify (motor)cyclists in a long-range (200 m), assuming quadratic-shaped detector pixels. Under the consideration of laser safety and using some basic assumptions of the variables in range equations, LiDAR systems using a blade irradiation fulfill the requirements with the shortest eye-safety distance (NOHD) of all considered system designs, which is 0.03 m in our example. For spot irradiation LiDAR systems, the maximum accessible emitted power is strongly limited by the high pulse frequency, reducing the detection range. Decreasing the pulse frequency can help obtain a more extended detection range under the premise of eye safety.

## Figures and Tables

**Figure 1 sensors-22-07532-f001:**
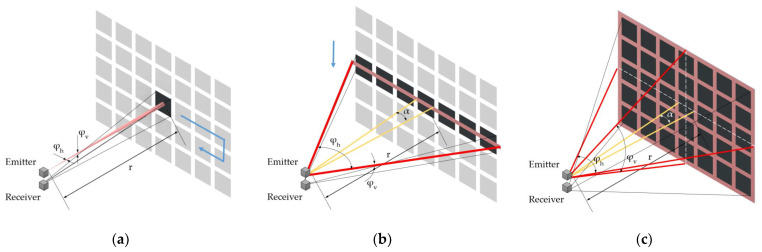
Radiation patterns of LiDAR systems according to [22]: (**a**) spot irradiation with collimated beam; (**b**) horizontal blade irradiation; and (**c**) flash irradiation of the entire FOV.

**Figure 2 sensors-22-07532-f002:**
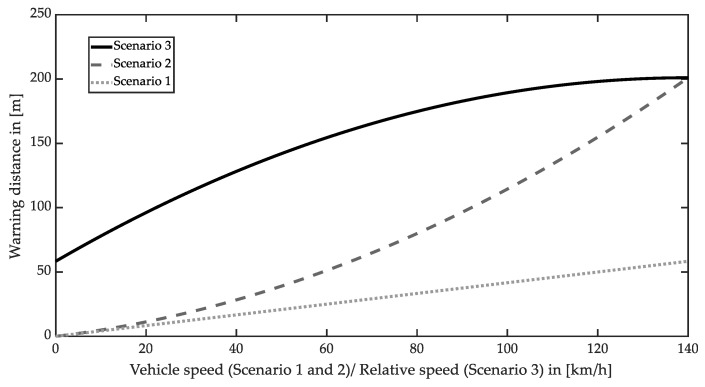
Warning distance as a function of the subject vehicle speed or relative speed for three typical scenarios; subject vehicle speed for scenario 3 is 140 km∙h^−1^.

**Figure 3 sensors-22-07532-f003:**
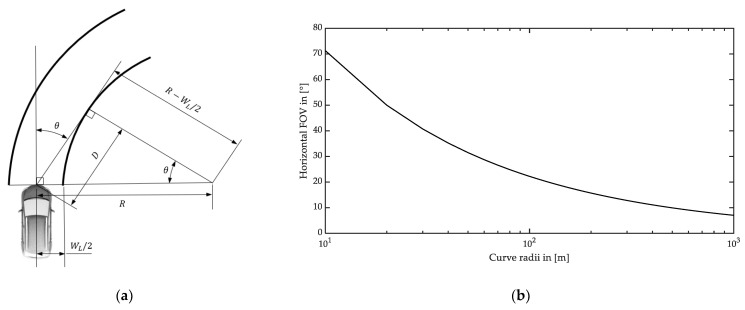
Detection area in a curve: (**a**) schematic to calculate the horizontal angle to cover the full lane width in a curve with radius R according to [29]; (**b**) required entire horizontal FOV for different curve radii.

**Figure 4 sensors-22-07532-f004:**
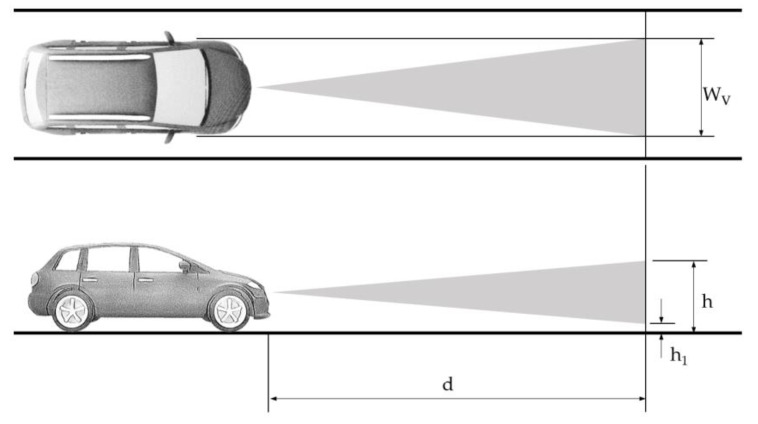
Determination of the FOV for the FVCWS function according to [29].

**Figure 5 sensors-22-07532-f005:**
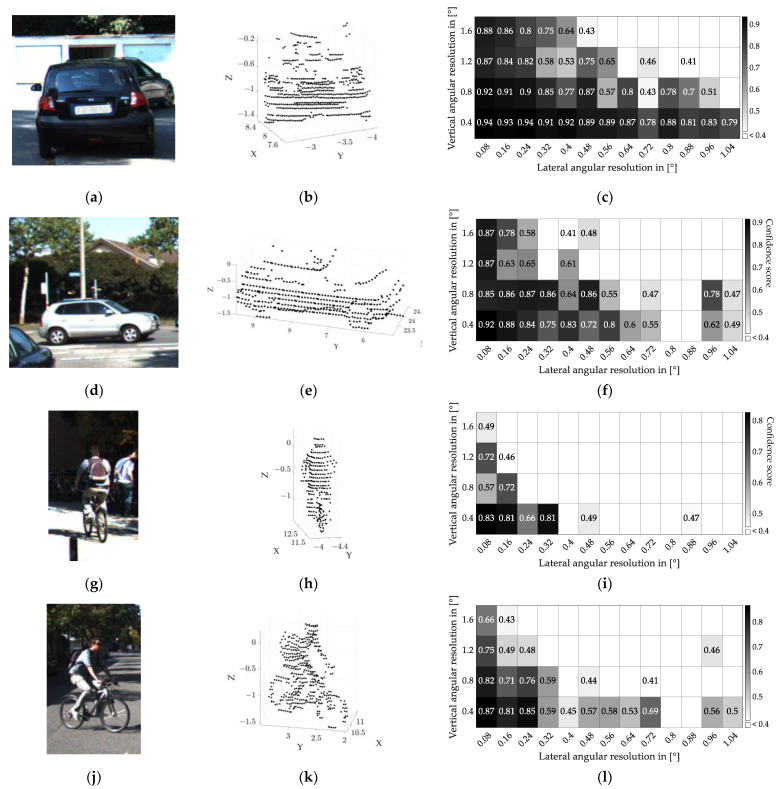
Detection results for cars, bicyclists, and pedestrians: (**a**–**c**) car in rear perspective; (**d**–**f**) car in lateral perspective; (**g**–**i**) bicyclist in rear perspective; (**j**–**l**) bicyblist in lateral perspective; (**m**–**o**) pedestrian. The left column shows RGB images of detection objects from the KITTI dataset [38]; the middle column shows raw point clouds of detection objects; and the right column shows the distribution of confidence scores varying with different angular resolutions.

**Figure 6 sensors-22-07532-f006:**
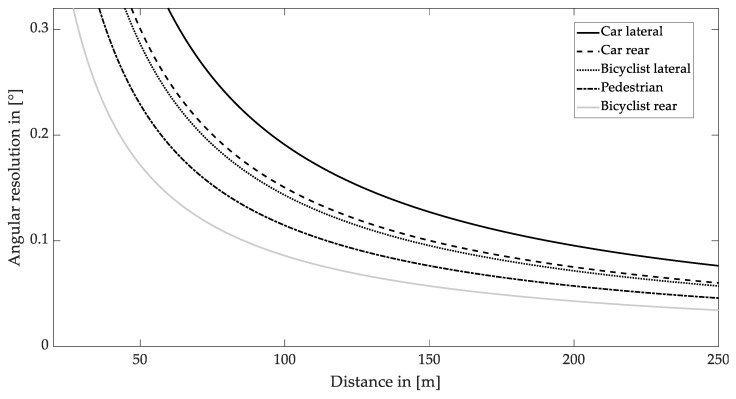
Required angular resolution for quadratic shaped pixels and different objects as a function of distance with a confidence score of 0.5.

**Figure 7 sensors-22-07532-f007:**
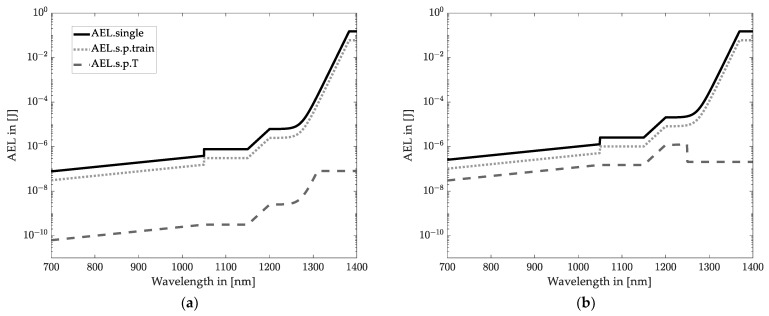
Maximum accessible emitting energy for (**a**) spot scanning LiDAR systems; (**b**) blade irradiation horizontal scanning LiDAR systems; (**c**) blade irradiation vertical scanning LiDAR systems; and (**d**) flash LiDAR systems.

**Figure 8 sensors-22-07532-f008:**
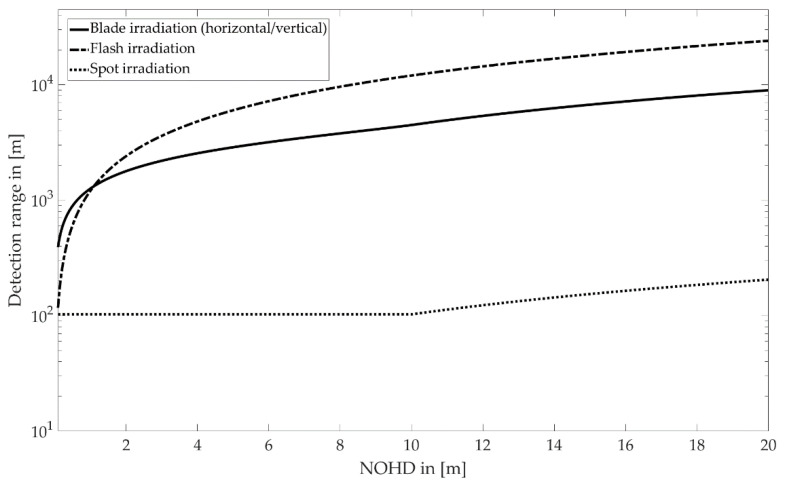
Detection range as a function of the eye safety distance for different LiDAR systems.

**Figure 9 sensors-22-07532-f009:**
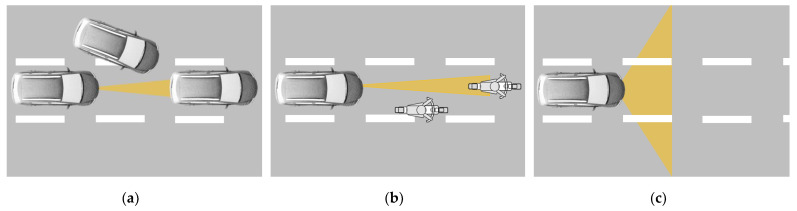
Illustration of exemplary problem situations due to a narrow FOV: (**a**) late detection of a cut-in vehicle; (**b**) offset of a motorbike; and (**c**) detection of three lanes.

**Table 1 sensors-22-07532-t001:** List of main variables in Equation (3).

Symbol	Quantity	Units
$P_{r}$	Received power	W
$P_{s}$	Emitted power	W
$T_{A}$	Atmospheric transmission	-
$\eta_{t}$	Optical efficiency of the emitter	-
$A_{s}$	Beam spread area of the emitter at the target	m^2^
Γ	Target cross-section	m^2^
ρ	Reflectance of the target	-
r	Distance between LiDAR and the target	m
$A_{R}$	Optical aperture of the receiver	m^2^
$\eta_{r}$	Optical efficiency of the receiver	-

**Table 2 sensors-22-07532-t002:** Range equations concerning a single pixel for various radiation patterns.

Case	Beam Spread Area $A_{s}$	Cross-Section Γ	Range Equation
Spot irradiation	$\pi \cdot {\tan(\varphi }_{h}/2)\cdot {\tan(\varphi }_{v}/2)\cdot r^{2}$	$\pi \cdot {\tan(\varphi }_{h}/2)\cdot {\tan(\varphi }_{v}/2)\cdot r^{2}$	$P_{r}{=P}_{s}\cdot K$
Blade irradiation	$4\cdot {\tan(\varphi }_{h}/2)\cdot {\tan(\varphi }_{v}/2)\cdot r^{2}$	$4\cdot \tan(\alpha /2)\cdot {\tan(\varphi }_{v}/2)\cdot r^{2}$	$P_{r}{=P}_{s}\cdot K\cdot \frac{\tan(\alpha /2)}{{\tan(\varphi }_{h}/2)}$
Flash irradiation	$4\cdot {\tan(\varphi }_{h}/2)\cdot {\tan(\varphi }_{v}/2)\cdot r^{2}$	$\Gamma_{3}=4\cdot {\tan(\alpha /2)}^{2}\cdot r^{2}$	$P_{r}{=P}_{s}\cdot K\cdot \frac{\tan(\alpha /2)}{{\tan(\varphi }_{h}/2)}\cdot \frac{\tan(\alpha /2)}{{\tan(\varphi }_{v}/2)}$

**Table 3 sensors-22-07532-t003:** Scenarios in FVCWS according to ISO 15623 [29].

No.	Scenario	Warning Distance
1	Preceding vehicle travels at ordinary speed	$v_{1}\cdot T$
2	Preceding vehicle is stationary	${\;v}_{1}\cdot {T+v}_{1}{}^{2}{/2a}_{1}$
3	Preceding vehicle decelerates with a relative speed $v_{\mathrm{rel}}$	${\;(T+v}_{\mathrm{rel}}{/a}_{1})\cdot v_{1}\;{–\;v}_{\mathrm{rel}}{}^{2}{/2a}_{1}$

**Table 4 sensors-22-07532-t004:** The required opening angle for FVCWS function with different curve radii according to [34].

Curve Class	Curve Radius	Full Horizontal FOV	Full Vertical FOV
Class Ⅰ	≥500 m	12.4°	5.2°
Class Ⅱ	≥250 m	18.0°	6.8°
Class Ⅲ	≥125 m	32.6°	10.2°

**Table 5 sensors-22-07532-t005:** Comparison of various 3D object detection algorithms based on the KITTI dataset.

Algorithms	Average Precision			Processing Time per Image (GPU/CPU)
	Cars	Pedestrians	Bicyclists	
PV-RCNN [39]	94.10%	66.38%	75.77%	0.1837 s/0.1726 s
PointRCNN [40]	92.90%	75.03%	76.76%	0.0861 s/0.0851 s
SECOND [41]	94.51%	71.94%	76.50%	0.0487 s/0.0488 s
PointPillars [42]	93.91%	65.46%	72.34%	0.0270 s/0.0286 s

**Table 6 sensors-22-07532-t006:** Minimum pixel number to detect main traffic targets under a confidence score of 0.5.

Object	Perspective	Aspect Ratio (B:H)	Min. Required Points	Min. Pixel Number
Car	Rear	2:1	31	8 × 4
	Lateral	4:1	25	12 × 3
Bicyclist	Rear	1:3	46	4 × 12
	Lateral	7:6	48	8 × 6
Pedestrian	-	3:8	14	3 × 8

## Data Availability

The data analyzed in this study are available via open access (KITTI dataset, https://www.cvlibs.net/datasets/kitti/ (accessed on 22 March 2022)). The downsampled version further analyzed here can be obtained from the authors, upon reasonable request.

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
