# Peer review of "Requirements for Automotive LiDAR Systems"

_sensors, 2022, doi:10.3390/s22197532_

Round 1
Reviewer 1 Report
The requirements of the detection range, field of view and angular resolution are essential for an automotive LiDAR system. This authors quantitatively elaborated these requirements in spot, blade, and flash radiation patterns in this manuscript respectively. Three functional principles such as TOF, AMCW, and FMCW are introduced firstly and the definitional equations of the detection range, field of view and angular resolution are presented. Followed by the quantitative analysis according to the different circumstances The relationship between laser safety and detection range are also discussed. Overall, this manuscript is well organized and written. My suggestions and comments are:
1) the error analysis should be added in each requirement calculation.
2) in equation (A2), the lower limit of integral theta should be -Pi/2.
Author Response
We have uploaded the file of the revised manuscript. Thank you very much for your constructive comments regarding our paper. Your suggestions have enabled us to improve our work.
best regards and have a great weekend.

Reviewer 2 Report
This study investigated the requirements for automotive LiDAR systems, such as range, detection quality, resolving power, field of view and eye safety. This work could contribute to evaluate automative context and their LiDAR systems. It is interesting and helpful for this topic research and industrial application.
However, there are some issues or questions could be addressed:
1) In the abstract, the experimental results and the important finds or value should be explained clearly.
2) In the Introduction of Section 1, it should involve the background and research aims clearly.
3) In the "State of the art" of Section 2, it should clearly indicate the existing problems within the previous LiDAR systems or methods.
4) In the Section 3 and Section 4, what is the important findings or contributions proposed by your studies? It should be explained clearly. Please also indicate and explain the innovation and research value clearly.
5) In Section 5, the authors should described their own findings in detail.
Author Response

(The authors gave the same response as above.)

Reviewer 3 Report
The paper describes the main requirements for automotive LiDAR systems. The details are illustrated and some helpful examples are provided; The topic of the paper is interesting for readers, and the paper has been well written and organized; following some minor comments will help for further improvement:
- The LiDAR system has already been studied in some papers; highlight the main contributions;
- Add a table to describe the main variables in the beginning;
-the quality of fig 2 is weak; revise it;
-fig 3. a has been designed in this paper? if not, refere it to the original paper;
-In Table 4, how the results are obtained; add some more details
-How equation 3 is simplified; add some more details;
-Add a direction for future studies using the type-3 fuzzy logic systems to tackle the uncertainties; such as: A type-3 logic fuzzy system: Optimized by a correntropy based Kalman filter with adaptive fuzzy kernel size
Author Response

(The authors gave the same response as above.)
